# From Micro-Perforates to Micro-Capillary Absorbers: Analysis of Their Broadband Absorption Performance through Modeling and Experiments

Cédric Maury [1] and Teresa Bravo [2,*]

1    Laboratoire de Mécanique et d'Acoustique (UMR), Centrale Marseille, Aix Marseille University, CNRS, 38 rue Frédéric Joliot-Curie, 13013 Marseille, France; cedric.maury@centrale-marseille.fr

2    Instituto de Tecnologías Físicas y de la Información (ITEFI), Consejo Superior de Investigaciones Científicas (CSIC), Serrano 144, 28006 Madrid, Spain

*    Correspondence: teresa.bravo@csic.es

**Featured Application: The design of compact low-frequency anechoic terminations.**

**Abstract:** A challenging issue is currently the design of non-fibrous ultra-thin acoustic absorbers that are able to provide broadband performance in demanding environments. The objective of this study is to compare using simulations and measurements the broadband absorption performance of highly porous micro-capillary plates (MCPs) to that of micro-perforated panels (MPPs) under normal incidence while considering unbacked or backed configurations. MCPs are unusual materials used for sound absorption with micron-sized channels and a high perforation ratio. Impedance-based modeling and Kundt tube experiments show that MCPs with suitable channel diameters have a pure constant resistance that outperforms the acoustic efficiency of MPP absorbers. Unbacked MCPs exhibit a controllable amount of high absorption that can exceed 0.8 over more than five octaves starting from 80 Hz, thereby achieving a highly sub-wavelength absorber. MCPs still provide broadband high absorption when backed by a rigid cavity. Their bandwidth-to-thickness ratio increases toward its causal limit when the cavity depth decreases. A parallel MCP resonant absorber partly backed by closed and open cavities is proposed. Such MCP-based absorbers could serve as short anechoic terminations for the characterization of acoustic materials at low frequencies.

**Keywords:** sound absorption; micro-perforated panels; micro-capillaries; anechoic termination

## 1. Introduction

Classical absorption and insulation materials present a limited performance in the low frequency range when imposing constraints on their total weight or thickness. However, ventilation or air conditioning units, as well as industrial machines such as compressors and turbines, present a preponderant low-frequency noise content. The use of micro-perforated panels (MPPs) [1] are well suited to such demanding environments as an alternative solution to porous or fibrous materials. Because they work on the principle of Helmholtz resonators, they can achieve remarkable absorption values but are limited to a narrow frequency band. To overcome this limitation, several solutions have been proposed to extend the performance of MPP resonance absorbers over a broader bandwidth toward the low frequency range.

The use of multi-layer and parallel partitions has been widely studied by the scientific community [2–5]. A pilot study was presented by Sakagami et al. [3] proposing the combination of two different MPPs arranged in parallel and periodically alternated. They provided a parametric study by varying the perforation ratios and depths of the backing cavities and calculated the excess attenuation provided. An array of three parallel-arranged MPPs with varying cavity depths was considered [4] to study how acoustic energy is

absorbed by the partition. A parallel absorption mechanism was effective assuming that the distance between the individual MPPs was smaller than a quarter-wavelength of the incident wave. The idea of merging different resonances for optimal sound absorption was further developed to explore the trade-off between sample dimensions, with a maximal constraint on the total thickness and the bandwidth of the absorption spectrum [5]. Using the causality principle, an absorber was designed composed of 16 quarter-wavelength coiled resonators, situated over a square section. With a depth of 59.3 mm, it was able to absorb noise above 345 Hz, provided that a 3 mm acoustic sponge covered the cavities. Another solution based on conventional resonators was proposed by Simon [6], who filled the backing cavity of a perforated plate with a set of tubes of different lengths. This generated an important shift in the absorption coefficient toward the low frequency range, allowing a reduction in the cavity depth.

To overcome the above limitations, a different approach is to consider unbacked MPPs, with panels working in free-standing conditions [7–10]. This configuration does not achieve resonant absorption, but it can provide notable values over a broad frequency range, provided that the physical parameters of the MPP are properly optimized. A selection of the hole diameters lower than 100 μm and an increase in the perforation ratio was found to be a requisite to obtaining broadband low-frequency absorbers [8]. Previously, ultra-microperforated panels (UMPPs), with perforation diameters of less than 80 μm, a perforation ratio below 19% and a backing cavity depth of 20 mm, provided wideband absorption that was suitable for a constrained space [9]. The drawback of this solution is the high cost of the manufacturing technique. Recently, MEMS technology was used in bio-microfluidics [10] and applied to silicon MPPs with micrometric holes [9]. However, the efficiency bandwidth of these solutions does not exceed two to three octaves.

The objective of the current work is to examine the acoustic performance of micro-capillary plates (MCPs) as ultra-broadband absorbers that are efficient in the low frequency range. The results are assessed against those provided by more standard materials, such as MPPs or UMPPs. Their potential applications include the design of compact low-frequency anechoic terminations. For instance, a 200 mm block of melamine foam with a conical shape was proposed as an anechoic termination to minimize reflections from the duct outlet [11]. It achieved a reflection coefficient ranging from 0.7 down to 0.05 when increasing the frequency. Unlike porous materials, MCPs can provide a controllable amount of constant absorption. It will be seen that they are tunable devices with the proper selection of their physical parameters.

Section 2 presents the morphology of MCPs and MPPs as well as their modeling and experimental characterization. In particular, an analytical formulation is provided for MCPs that differs from the classical continuous laws applied to MPPs. These models are used in Section 3 to outline the acoustic properties of unbacked MCPs and MPPs covering hole diameters from micrometric to millimetric sizes. The simulation results are validated through an experimental set-up situated in a semi-anechoic chamber. The acoustic performance of MCPs and MPPs when backed by a closed or an open cavity is examined in Section 4 using simulations and measurements. A comparison between the broadband acoustic efficiencies of MCPs and MPPs is discussed in Section 5 to better understand the advantages and limitations of the absorbers. This leads to the proposal of an ultra-short broadband anechoic termination.

## 2. Materials, Modeling and Characterization

### 2.1. Materials

A number of thin micro-porous materials, ranging from MPPs with sub-millimetric circular holes to MCPs with micron-sized cylindrical channels, are described in Table 1 according to the diameter $d$ of their holes, the center-to-center distance $\Lambda$ between the holes (or holes pitch), their perforation ratio $\sigma$ and their thickness $t$. Magnified views of their morphology are shown in Figure 1 (resp. Figure 2) for the micro-capillaries of MCP1 (resp. MCP2). Figure 2 also shows the micro-perforations of a UMPP and MPP. Two groups of

materials are distinguished: highly porous materials (MCP1 and MCP2) with a perforation ratio greater than 50% and more standard materials with a porosity lower than 10% (UMPPs and MPPs). MCP1 and MCP2 are composed of 113,000 (resp. 1600) channels per cm$^2$ for MCP1 (resp. MCP2), as seen in Figure 1, resulting in a perforation ratio exceeding 50% for both materials. By comparison, the UMPP and MPP only have 25 holes per cm$^2$, as can be seen in Figure 2b,c. In the first group, MCP1 is an innovative material with both a high porosity and micron-sized channels, whereas MCP2 is a highly porous material with only sub-millimetric sized channels (0.2 mm). It can be seen in Sections 2.2 and 3 that MCP1 has a pure calibrated resistance with a negligible "effective mass" effect [1]. In the second group, the size of UMPP apertures (0.3 mm) is smaller than that of MPP holes (0.88 mm), but they are comparable to that of MCP2 channels with, however, a much scarcer distribution of the apertures. This results in an increasing "effective mass" effect (defined in Section 2.2) for MCP2, the MPP and the UMPP, which limits the broadband performance of the absorber, as verified in Section 4.

**Table 1.** Geometric parameters of the micro-porous panels.

| Material Acronym | Hole Diameter (µm) | Hole Pitch (µm) | Perforation Ratio (%) | Thickness (mm) |
|---|---|---|---|---|
| MCP1 | 28 | 32 | 69.4 | 1 |
| MCP2 | 200 | 268 | 50.5 | 1 |
| UMPP | 300 | 3000 | 1.8 | 0.5 |
| MPP | 880 | 3000 | 6.8 | 1 |

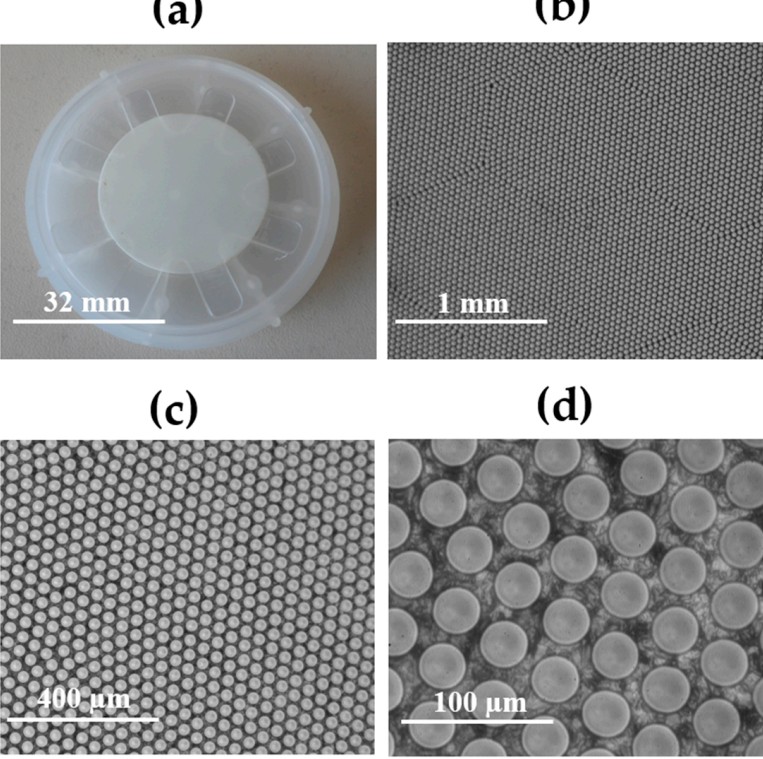

**Figure 1.** (**a**) Photograph of MCP1 with an outer diameter of 35 mm and thickness of 1 mm; (**b**) Magnification (factor 4) of MCP1 revealing a hexagonal distribution for the set of micro-channels; (**c**) Magnification (factor 10) of MCP1 showing micro-channel staggered alignments; (**d**) Magnification (factor 40) of MCP1 showing 60° staggered cylindrical micro-channels with diameters of 28 µm and hole pitches of 32 µm.

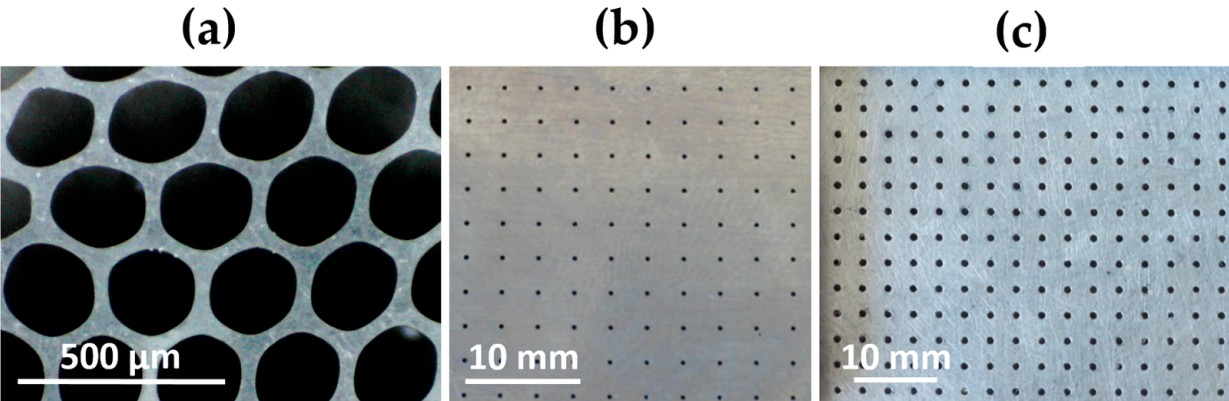

**Figure 2.** (**a**) Magnification (factor 10) of MCP2 revealing 60° staggered near-circular micro-channels with diameters of 200 μm and hole pitches of 268 μm; (**b**) Photograph of UMPP showing aligned distributions of circular holes with diameters of 0.3 mm and hole pitches of 3 mm; (**c**) Photograph of MPP showing aligned distributions of circular holes with diameters of 0.88 mm and hole pitches of 3 mm.

Figures 1d and 2a show that MCP1 and MCP2 are composed of 60° staggered circular channels so that their perforation ratio is calculated as $\sigma = \pi d^2/\left(2\sqrt{3}\Lambda^2\right)$, whereas the UMPP and MPP are composed of aligned circular holes so that $\sigma = \pi d^2/(4\Lambda^2)$. The UMPP and MPP are manufactured using micro-machining stainless steel and aluminum panels, respectively. MCP1 and MCP2, composed of leaded glass, are produced via an acid-etching process. Soluble fibers are inserted into an insoluble glass matrix that is melted and stretched. After slicing to the desired thickness, the plates are placed in an acid bath to solve the fibers, hence producing a porous solid with a high density of micro-channels per cm². Usually, MCPs are used to enhance the resolution of optical sensors such as cosmic ray intensifiers. On top of their channel density, another distinct feature of MCPs compared to MPPs is their high thickness-to-diameter ratio, $t/d$, that reaches values of 37 for MCP1 and 5 for MCP2, whereas it has values of 1.7 for UMPPs and 1.1 for MPPs.

*2.2. Modeling*

The first step is to calculate the transfer impedance $Z$ of a circular micro-aperture of radius $a$, assimilated to a short cylindrical tube whose inner axial particle velocity $\bar{v}$, averaged over the tube cross-sectional area, is driven by the pressure difference $\Delta p$, with harmonic time-dependence ($e^{j\omega t}$), exerted on either side of the tube, so that $\Delta p = Z\bar{v}$. Following Maa's derivation of $Z$ for micro-perforations filled with a viscous fluid [1], $v$ satisfies the linearized equation of momentum conservation in cylindrical coordinates:

$$\frac{1}{r}\frac{\partial}{\partial r}\left(r\frac{\partial v}{\partial r}\right) - j\left(\frac{\mathrm{Sh}}{a}\right)^2 v = \frac{1}{\mu}\frac{\Delta p}{t}, \tag{1}$$

which is expressed in terms of the Shear number $\mathrm{Sh} = a/\delta_v$ and the ratio between the aperture radius and the thickness $\delta_v = \sqrt{\mu/(\omega\rho_0)}$ of the viscous boundary layer, with $\mu$ as the air dynamic viscosity, $\rho_0$ as the air density and $\omega$ as the angular frequency. Depending on the values of the Knudsen number, $\mathrm{Kn} = \widetilde{\Lambda}/a$, the ratio between the molecular mean-free path of air particles ($\widetilde{\Lambda} = 68$ nm under normal conditions of pressure and temperature) and the aperture radius $a$, different boundary conditions are fulfilled over the aperture wall surface $\Sigma$. If $\mathrm{Kn} \leq 10^{-3}$, which is the case for MCP2, UMPPs and MPPs, continuum laws stay valid, and a no-slip boundary condition is satisfied.

$$v_\Sigma = 0, \tag{2}$$

which is fulfilled over the aperture walls ($r = a$). If $10^{-3} \leq \mathrm{Kn} \leq 10^{-1}$, which is the case for MCP1, a slip-flow regime holds due to tangential momentum exerted on the aperture walls [12]. Equation (2) is then replaced by the following boundary condition [13]:

$$v_{\Sigma} = B_v \frac{\partial v_{\Sigma}}{\partial r},\tag{3}$$

which is satisfied over the aperture walls, with $B_v = \mathrm{Kn}(2 - \sigma_v)/\sigma_v$ expressed in terms of $\sigma_v$, the tangential momentum accommodation coefficient, which is equal to 0.9 for an air–glass interface [14].

Solving Equations (1) and (2) for $v$, dividing by the perforation ratio and adding outer correction terms lead to the effective transfer impedance for an MPP [1]:

$$Z_{\mathrm{MPP}} = \frac{\mu\,\mathrm{Sh}}{\sqrt{2}a\,\sigma} + \mathrm{j}\omega\frac{\rho_0\,t}{\sigma}\left\{\left[1 - F\left(\mathrm{Sh}\sqrt{-\mathrm{j}}\right)\right]^{-1} + \frac{\delta\,a}{t}\right\},\tag{4}$$

which is expressed in terms of $F(x) = 2J_1(x)/[x\,J_0(x)]$ with the $J_1$ and $J_0$ Bessel functions of the first kind of orders 1 and 0, respectively. The inner term, $\mathrm{j}\omega\,\rho_0\,t\left[1 - F\left(\mathrm{Sh}\sqrt{-\mathrm{j}}\right)\right]^{-1}\sigma^{-1}$, accounts for viscous dissipation and inertial effects inside the holes, whereas the outer terms, $\mu\,\mathrm{Sh}/\left(\sqrt{2}a\,\sigma\right)$ and $\mathrm{j}\omega\,\rho_0\delta\,a/\sigma$, respectively account for outer frictional losses and air motion inertial effects at the inlet/outlet of the holes, with $\delta = 16/(3\pi)$, the added-length correction factor [1]. An "effective mass" per unit area, $\rho_0(t + \delta\,a)/\sigma$, is defined using Equation (4), which steadily increases for MCP2, MPPs and UMPPs, but that is negligible for MCP1.

Solving Equations (1) and (3) for $v$, dividing by the perforation ratio and assuming a narrow channel behavior, $\mathrm{Sh} < 1$, provide the following expression for the effective transfer impedance of an MCP with a micrometric channel diameter [8]:

$$Z_{\mathrm{MCP}} = \frac{8t\mu}{\sigma a^2(1 - 4B_v)} + \mathrm{j}\omega\frac{4\rho_0\,t}{3\sigma} \approx \frac{8t\mu}{\sigma a^2(1 - 4B_v)}.\tag{5}$$

It reduces to a pure acoustic resistance, with a constant value over a broad frequency range, and minute reactance due to a high perforation ratio, as illustrated in Figure 3a,b, respectively. Unlike MPPs, the outer correction terms of MCPs can be neglected due to their micrometric channel diameters and large perforation ratios. Equation (5) shows that the constant resistance of an MCP is controlled by the channel radius following an inverse-power law dependence with an order of 4. Conversely, Figure 3 shows that MPPs exhibit a much lower resistance than MCP1 as well as a greater reactance, which both increase with the frequency. Note that Equations (4) and (5) describing the effective transfer impedance of MCPs and MPPs assume a sufficient number of holes/channels per acoustic wavelength $\lambda$, typically $\Lambda < \lambda/4$, which is clearly verified for MCPs.

Different load conditions can be applied behind the porous materials. In unbacked configurations, a simple model assumes a plane wave anechoic load, such that the overall input impedance over the material surface is given by $Z_{\mathrm{in}} = Z_{\mathrm{MPP,MCP}} + Z_0$ with $Z_{\mathrm{MPP,MCP}}$ given by Equations (4) or (5) and $Z_0 = \rho_0 c_0$, the plane wave impedance. From this model, Figure 3c shows that MCP1 is more dissipative than an MPP, which is essentially reactive over the whole frequency range.

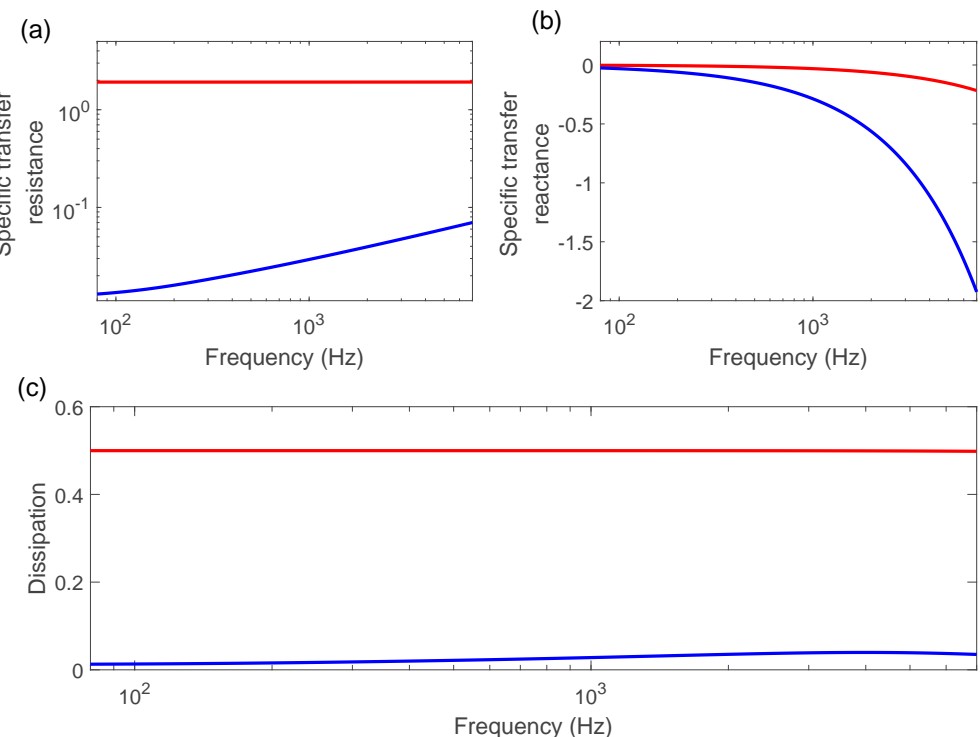

**Figure 3.** (**a**) Simulated specific transfer resistance, $\text{Real}(Z_{\text{MPP,MCP}}/Z_0)$; (**b**) Simulated specific transfer reactance, $\text{Imag}(Z_{\text{MPP,MCP}}/Z_0)$; (**c**) Simulated dissipation of MCP1 (red) and MPP (blue) assuming normal incidence and plane wave anechoic load; $Z_{\text{MPP}}$ and $Z_{\text{MCP}}$ are given by Equations (4) and (5), respectively.

In practice, porous samples may be inserted at the open end of a Kundt tube to achieve an anechoic termination, as described in Section 2.3. Therefore, a cost-efficient realistic model of the radiation impedance of a flanged termination is required. The model established by Silva et al. [15] is used in the unbacked configuration. It has been verified using measurements behind micro-porous materials (see Section 3), assumed to be circular with radius $R$. The radiation impedance is then modeled as $Z_{\text{RAD}} = (1 + R_{\text{RAD}})(1 - R_{\text{RAD}})^{-1}$ with

$$R_{\text{RAD}} = -\left(1 + \text{j}\frac{k_0 R}{c_1}\right)^{-(c_2+1)}, \tag{6}$$

$k_0 = \omega/c_0$, $c_1 = 0.8216$ and $c_2 = 0.35$. The input impedance of the unbacked materials is then calculated as $Z_{\text{in}} = Z_{\text{MPP,MCP}} + Z_{\text{RAD}}$.

Backing the porous samples with a rigidly closed cavity with depth $D$ leads to resonant absorbers whose input impedance is given by

$$Z_{\text{in}} = Z_{\text{MPP,MCP}} + \text{j}Z_0 \cot\left(\widetilde{k}_{0,R} D\right), \tag{7}$$

with $\widetilde{k}_{0,R}$, a complex wavenumber that describes viscothermal attenuation effects over the cavity walls [8]. The imaginary part of $Z_{\text{in}}$ vanishes at the resonance frequencies of the absorber. In the case of a rigidly backed MPP, the first zero is associated with a Helmholtz-type hole–cavity resonance that occurs at

$$f_{\text{HC,MPP}} = \frac{1}{2\pi}\sqrt{\frac{\rho_0 c_0^2}{mD}}, \tag{8}$$

with $m \approx \rho_0 t / \sigma$, the MPP effective mass. It is followed by high-order quarter-wavelength resonances downshifted by a large effective mass due to a low perforation ratio. In the case of a rigidly backed MCP, all the resonances are of quarter-wavelength types that occur at

$$f_{n,\text{MCP}} = \frac{(2n+1)\,c_0}{4D},\tag{9}$$

with $n$, an integer. The absorption peaks are associated with the maximum particle velocity at these resonances over the MPP or MCP samples. The absorption is obtained as the fraction of the incident energy that is not being reflected. It reads $1 - |r|^2$ with $r = (Z_{\text{in}} - Z_0)/(Z_{\text{in}} + Z_0)$, the reflection coefficient, and $Z_{\text{in}}$, given by Equation (7). In the rigidly backed case, it coincides with the dissipation, whereas in the unbacked case, as is the case in Figure 3c, the dissipation is the fraction of the incident energy that is neither reflected nor transmitted. It is calculated as $1 - |r|^2 - |\tau|^2$ with $\tau = 2Z_0/(Z_{\text{in}} + Z_0)$, the transmission coefficient.

### 2.3. Characterization

The validity of the previous models, especially of the MCP model (Equation (5)), is assessed against absorption measurements carried out using a small impedance tube mounted vertically and located in a semi-anechoic room, as shown in Figure 4, for the testing of the unbacked and backed porous absorbers. The absorption is measured using the two-microphone transfer function method following the normative ISO-10534-2 [16]. It uses a Kundt tube that is 800 mm long with a radius of $R = 15$ mm and 10 mm-thick walls made of steel. Its base is connected to a loudspeaker that generates a pressure field composed of plane waves up to the first duct cut-on frequency, $f_{\text{cut-on}} = 1.84c_0/(2\pi R) = 6700$ Hz. At the top of the tube is an open termination onto which the circular porous samples are inserted (Figure 4 left). The samples can be backed by a closed cavity with a varying depth (Figure 4 right). The piston on top of the closed cavity may be removed to achieve an open cavity with a depth of 152 mm. Therefore, three configurations can be tested: unbacked, backed by closed cavities with a varying depth or backed by an open cavity with a fixed depth.

A white noise signal drives the loudspeaker between 80 Hz and 7 kHz. According to ISO-10534-2, a transfer function $H_{12}$ is evaluated between a pair of 1/4″ condenser microphones, flush-mounted on a tube wall, separated by a distance of $d_m = 50$ mm and located at a distance of $e_m = 67$ mm apart from the tube termination. The distance between the microphones is lower than the smallest half-wavelength at the duct cut-on frequency. Moreover, above 80 Hz, the smallest detectable phase difference over an acoustic wavelength is greater than the phase mismatch between the microphones. During the measurement, the output signals from the microphones are acquired using the OROS (type OR38) multi-channel system over a bandwidth of 80 Hz–6.7 kHz, at a sampling rate of 12.8 kHz and with a spectral resolution of 1.56 Hz, triggered by the generation of the drive signal. The reflection coefficient over the tested sample is then calculated as

$$r = e^{-2jk_0(d_m+e_m)}\frac{H_{12} - e^{-jk_0 d_m}}{e^{jk_0 d_m} - H_{12}},\tag{10}$$

from which the absorption coefficient $1 - |r|^2$ is readily obtained.

Near-field measurements of the sound field radiated by the unbacked samples are performed using a calibrated pressure–velocity (p–v) probe [17] located at a short stand-off distance (10 mm) from the porous samples, as shown in Figure 4 left. This distance avoids the occurrence of pressure or velocity nodes below 6.7 kHz while keeping a signal-to-noise ratio greater than 20 dB. The p–v probe enables collocated measurements of both the pressure and the three components of the radiated acoustic velocity. It can serve to estimate the radiation impedance of the samples. It can be displaced along a diameter of the samples to evaluate the directivity of the radiated intensity.

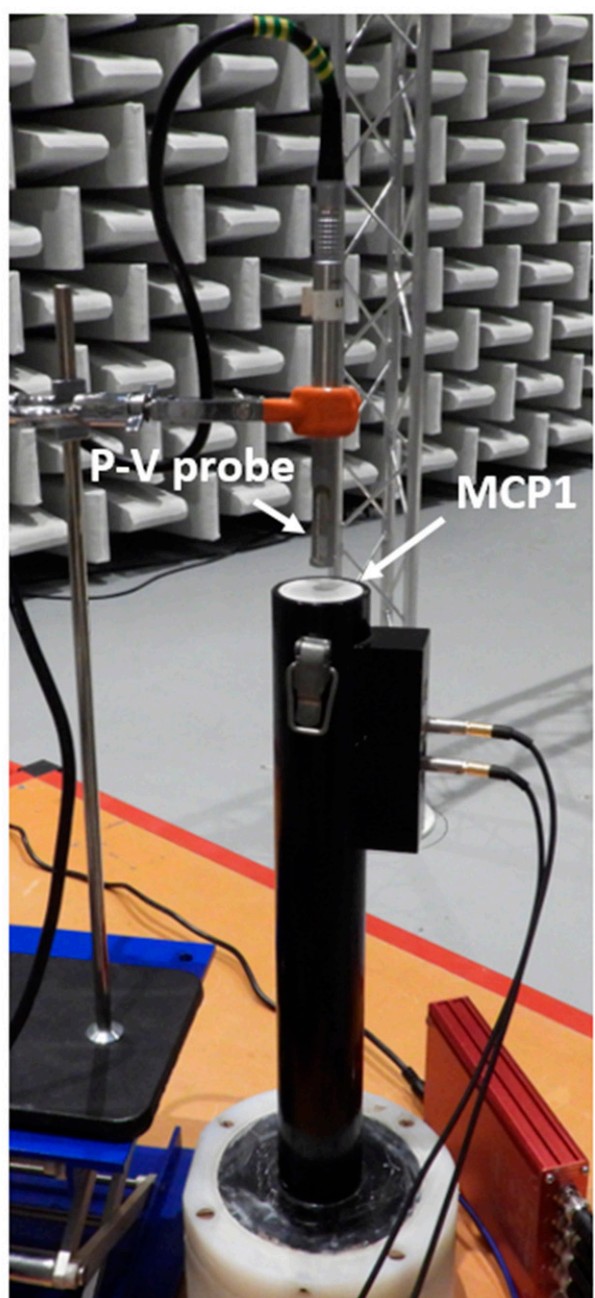 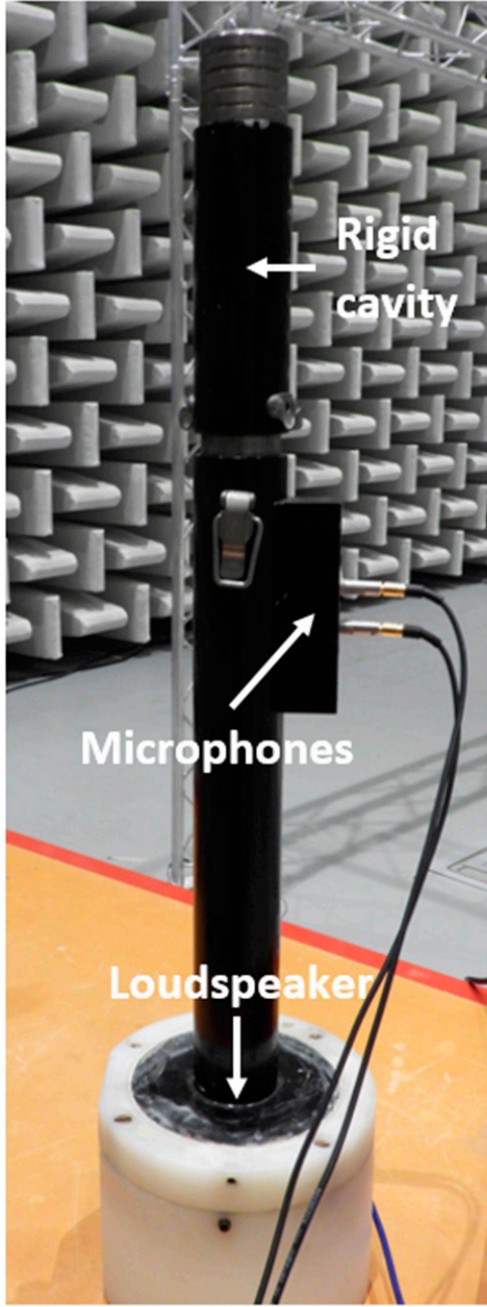

**Figure 4.** Photographs of the Kundt tube used to measure the absorption properties of an unbacked (**left**) and rigidly backed (**right**) MCP in an anechoic environment; a pressure–velocity (p–v) probe characterizes the acoustic radiation of the unbacked MCP.

## 3. Acoustic Properties of Unbacked MCPs and MPPs

Simulations and measurements were carried out to determine the absorption properties of the thin porous materials listed in Table 1 when unbacked. As shown in Figure 4 on the left, the material samples were inserted at the termination of a Kundt tube, thereby achieving an ultra-thin termination with a millimetric thickness. Their measured absorption properties are shown in Figure 5. Their unbacked transmitting side radiates in an anechoic environment. The radiation properties (impedance and acoustic intensity) of the samples, characterized by a 3D pressure–velocity probe described in Section 2.3, are shown in Figures 6 and 7.

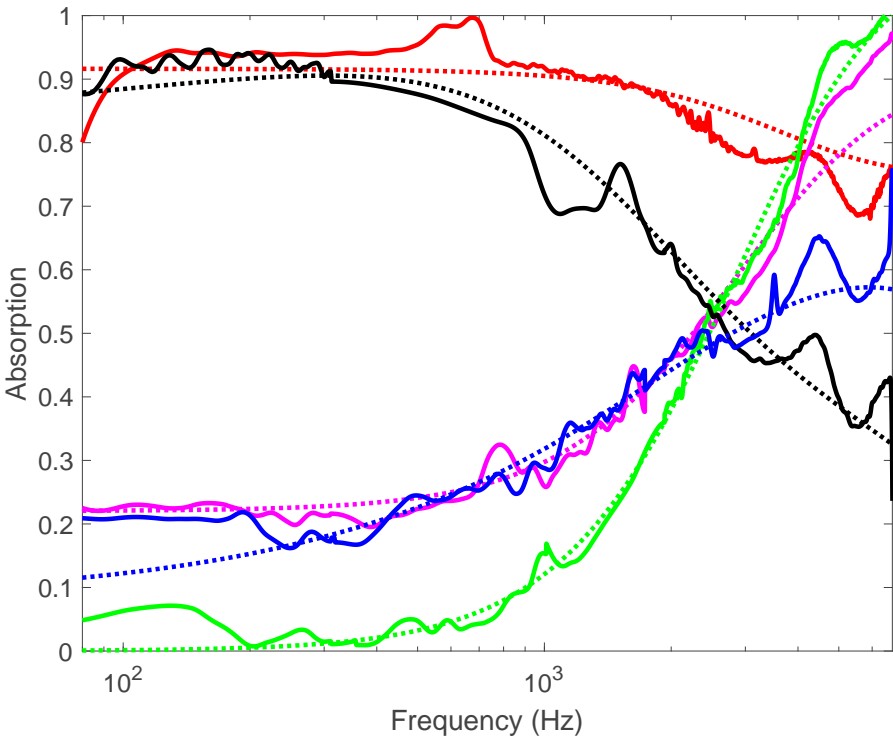

**Figure 5.** Simulated (dotted) and measured (plain) absorption properties of unbacked MCP1 (red), MCP2 (magenta), UMPP (black), MPP (blue) and open termination (green) assuming flanged radiation conditions and normal incidence.

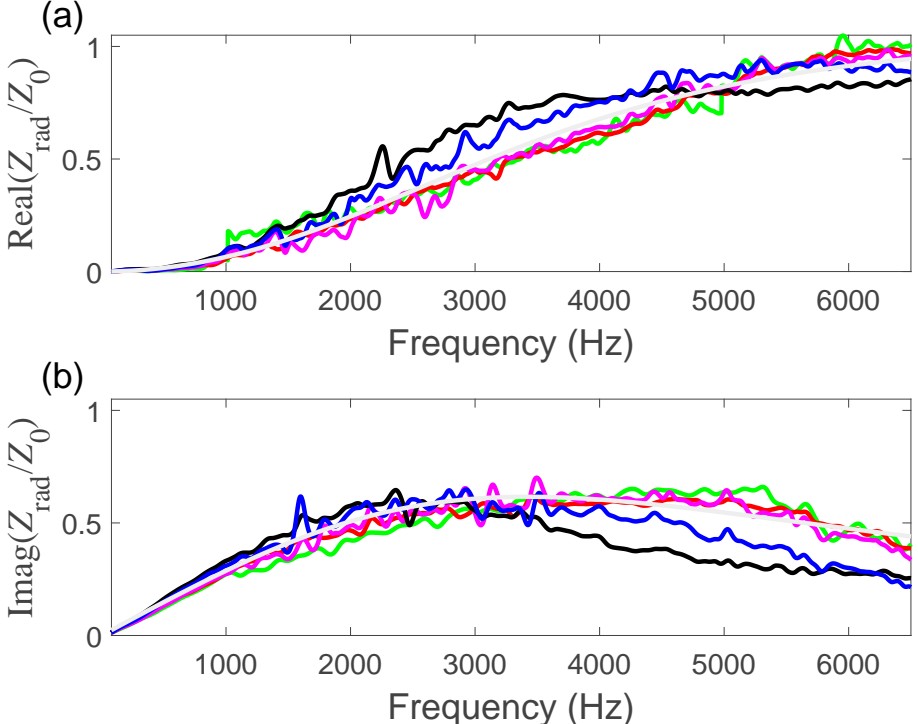

**Figure 6.** Specific radiation resistance (**a**) and reactance (**b**) of MCP1 (red), MCP2 (magenta), UMPP (black) and MPP (blue) and open termination (green) measured with a p–v probe compared with a radiation impedance model (gray) assuming a flanged open termination.

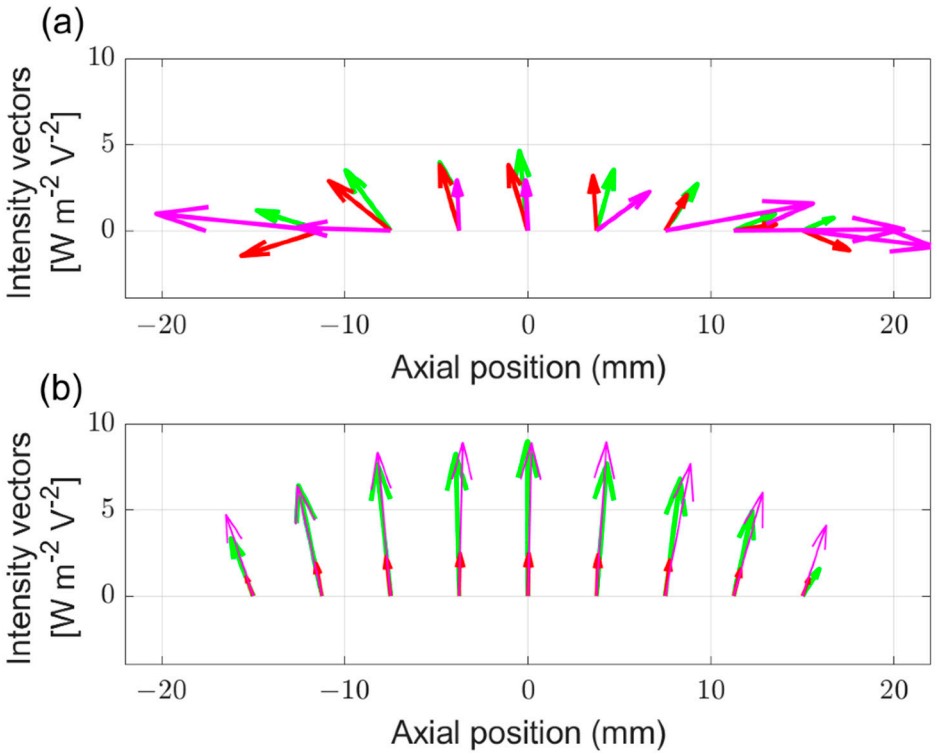

**Figure 7.** Measured intensity vectors at 300 Hz (**a**) and at 5000 Hz (**b**) associated with the acoustic radiation of an open termination (green), MCP1 (red) and MCP2 (magenta).

It can be seen from the green curve of Figure 5 that the open termination (reference case) reflects a large amount of the incident energy below 1000 Hz (more than 90%). As the frequency increases, part of the incident energy transmitted by the open outlet gradually rises, reaching unity at 6500 Hz. As shown in Figure 6 (green and gray curves), this open tube behavior is well predicted by the radiation impedance model of Silva et al. [15] described by Equation (6) for a flanged cylindrical termination.

It is remarkable to observe that closing the open termination of MCP1 considerably enhances the absorption above 0.9 over a broad bandwidth of between 100 Hz and 1200 Hz (Figure 5 (plain red curve)). MCP1 thus achieves a highly absorbing termination with a deep sub-wavelength thickness $0.0003\lambda$ (resp. $0.004\lambda$) at 100 Hz (resp. at 1200 Hz) with $\lambda$, the acoustic wavelength. Over this wide frequency range that covers almost four octaves, high absorption values are essentially due to visco-thermal dissipation effects through the micron-sized apertures of MCP1. Above 1200 Hz, the absorption slightly decreases down to 0.75 at 6500 Hz due to the progressive decay of the reactance (Figure 3b (red curve)). This plateau-and-decay behavior observed in the measured absorption spectrum of MCP1 is well predicted by the impedance model (Figure 5 (dotted red curve)), assuming that the radiation load impedance behind MCP1 is that of a flanged termination. This assumption is verified when comparing the red and gray curves of Figure 6. They show that the radiation impedance measured behind the highly porous MCP1 material closely follows the impedance of a flanged termination.

Like MCP1, MCP2 belongs to a class of highly porous materials with a perforation ratio greater than 50%. However, unlike MCP1, which essentially works on visco-thermal dissipative effects, the apertures of MCP2 are seven times larger than those of MCP1. Therefore, MCP2 has a lower resistance. It is much less dissipative and more transmissive than MCP1. Above 2 kHz, it behaves like an acoustically transparent open termination, as shown by the close proximity between the green and magenta curves in Figure 5. At low frequencies, the low absorption values of MCP2 (around 0.2) are due to moderate dissipation and transmission through the channels of MCP2.

The UMPP and MPP have perforation ratios that are considerably lower than those of MCP1 and MCP2, so they have much larger reactance values than those of MCP1 and MCP2 as the frequency increases. This is verified in Figure 3b when comparing the reactance of MCP1 to that of the MPP. Hence, the UMPP and MPP tend to be reflective at high frequencies, especially the UMPP, whose absorption curve already decays from 400 Hz down to 0.4 at 6500 Hz (Figure 5 (black curve)). Acoustic reflections on the UMPP are more pronounced than those on the MPP due to the perforation ratio of the UMPP that is about four times lower than that of the MPP, as well as due to smaller hole diameters.

At low frequencies, it is of interest to observe that UMPPs achieve absorption values that are similar to those of MCP1 (around 0.9) but over a smaller bandwidth (only two octaves) that ranges between 100 Hz and 400 Hz. A larger acoustic flow distortion [18] occurs at the inlet/outlet of the holes of UMPPs with respect to the MPPs due to a hole pitch that is larger than that of MPPs. Hence, the resistance of UMPPs is 15 times greater than that of MPPs. It therefore produces a sufficient amount of dissipation and transmission to achieve high absorption values up to 400 Hz, a frequency at which the viscous boundary layer (VBL) occupies half of the radius of the holes of UMPPs. Beyond 400 Hz, the VBL thickness further decreases, and the dissipative effects tend to be overwhelmed by the reflection so that the frequency increases.

On the transmitting side, it can be seen from Figure 6 that, amongst the four materials, only those with the lowest porosity, the UMPP and MPP, exhibit the largest discrepancy (up to 20%) with respect to the radiation impedance of a flanged open termination. However, these differences do not drastically impede the correlation between the simulated and measured absorptions, as can be seen in Figure 5. Assuming rigid panels, the sound field radiated by the materials is the result of the sound fields radiated either by a discrete number of apertures, e.g., by 25 holes per cm$^2$ for the UMPP and MPP, or by a continuum of apertures, e.g., by thousands of parallel micro-channels per cm$^2$ for MCP1 and MCP2. In the latter case, the high porosity and unit acoustic flow tortuosity across the channels explain that one can assimilate the radiation of micro-capillary materials to that of an open flanged termination.

Figure 7 shows how the highly porous materials (MCP1 and MCP2) modify the radiation properties of the open termination (green arrows), which is highly reflective at low frequencies (Figure 7a) and highly transmissive at high frequencies (Figure 7b). At high frequencies, the intensity vectors are essentially parallel and point out toward a direction that is normal to the open termination (Figure 7b). This corresponds to plane wave propagation conditions where pressure and axial velocity are in-phase. At low frequencies, monopole-type radiation conditions induce tangential intensity components that are preponderant toward the edge of the open termination (Figure 7a). This is due to out-of-phase relationships between the pressure and the axial velocity that are associated with near-field effects.

Closing the open termination for MCP1 or MCP2 has rather distinctive effects. MCP1 (red arrows) is highly dissipative over the whole frequency range. At low frequencies, it therefore blocks sound transmission (similar to the reflective open termination). At high frequencies, the dissipative effects of MCP1 also reduce sound transmission. This effect is clearly seen in Figure 7b with a lower amplitude of the intensity vectors of MCP1 with respect to those of the transmissive open termination. At low frequencies, MCP2 (magenta arrows) is more transmissive than MCP1 and the open termination. This results in larger amplitudes of the radiated intensity vectors, especially toward the grazing angles, as can be seen in Figure 7a. At high frequencies (typically above 2000 Hz), MCP2 transmits as much sound as the open termination. This results in intensity vectors with similar amplitudes (Figure 7b) and a similar amount of absorption above 2000 Hz (Figure 5) with respect to the open termination.

Wideband high absorption performance is thus observed for non-resonant absorbers, such as the MCP1 and UMPP unbacked materials. They are well predicted by the simulation models of Section 2.2. In practice, a backing cavity is necessary to achieve resonant

absorbers. The next section examines the absorption performance of the four materials when backed by closed or open cavities.

## 4. Acoustic Properties of MCPs and MPPs Resonant Absorbers

Measurements were carried out for the absorption performance of the four types of materials listed in Table 1 when backed by a rigid cavity with a variable depth, as illustrated in Figure 4 (right). Table 2 summarizes the absorption performance of the individual resonators with $\bar{\alpha}$, the total absorption evaluated between 80 Hz and $f_{\text{cut-on}}$; $\alpha_{\text{max.}}$, the value of the first absorption peak; and $f_1$, $f_2$, the lower and upper frequencies delimiting the bandwidth at 50% of the maximum of the first absorption peak. Figure 8 presents the measured and simulated absorption results, assuming a cavity depth of 55 mm.

**Table 2.** Measured absorption performance of individual resonators. Bold values represent the best performance for each indicator.

| Material—Cavity Depth (mm)—Backing Type | $[f_1 - f_2]$ (Hz) | $\alpha_{\text{max.}}$ | $\bar{\alpha}$ |
|---|---|---|---|
| MCP1—55—Closed | 300–3000 | 0.92 | 0.71 |
| MCP1—35—Closed | 435–4800 | 0.91 | 0.70 |
| MCP1—25—Closed | **480–6500** | 0.86 | **0.76** |
| MCP1—20—Closed | 512–6500 | 0.87 | 0.75 |
| MCP1—10—Closed | 700–6500 | 0.82 | 0.69 |
| MCP1—152—Open | 85–530 | 0.83 | 0.73 |
| MCP2—55—Closed | 540–2760 | 0.48 | 0.45 |
| MCP2—152—Open | 80–520 | 0.43 | 0.73 |
| UMPP—55—Closed | 330–2160 | **0.99** | 0.51 |
| UMPP—20—Closed | 800–3000 | **0.99** | 0.54 |
| UMPP—152—Open | 80–526 | 0.97 | 0.55 |
| MPP—55—Closed | 730–2300 | 0.40 | 0.42 |
| MPP−152−Open | 80–520 | 0.22 | 0.58 |

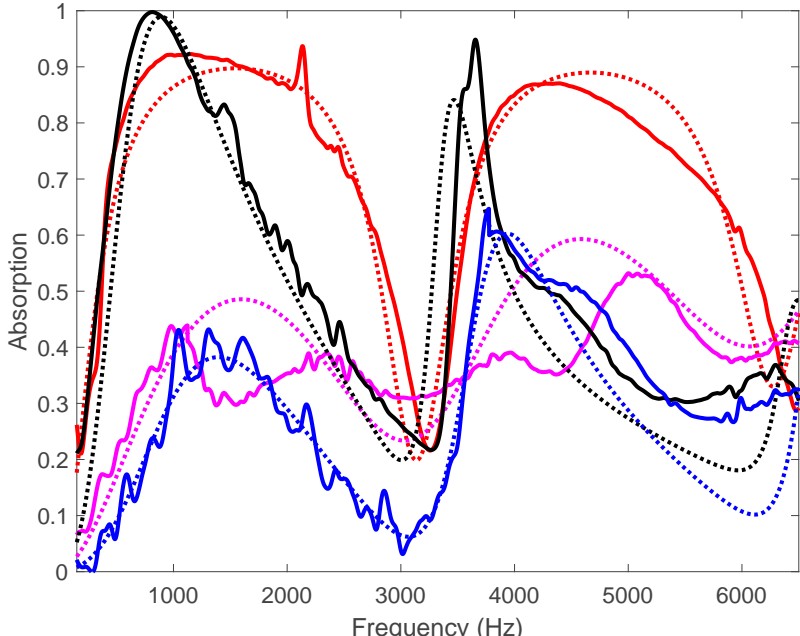

**Figure 8.** Simulated (dotted) and measured (plain) absorption properties of MCP1 (red), MCP2 (magenta), UMPP (black) and MPP (blue) backed by a rigid cavity with a depth of 55 mm, assuming normal incidence.

Both the simulation and measurements show the effects of the two first acoustic resonances on the absorption performance of the rigidly backed materials up to 6500 Hz. The

highly dissipative MCP1 provides the greatest performance for a 55 mm cavity depth with a total frequency-averaged absorption of 0.71 up to 6500 Hz, with maximum absorption values of up to 0.92 and with the broadest bandwidth that extends to 50% of the maximum over a range of 2700 Hz around each resonance (see Table 2). The bandwidth of MCP1 resonances spans three octaves.

The performance of MCP2, which is less dissipative, is more moderate than that of MCP1 with a total absorption of 0.45, a maximum absorption of 0.48 and a bandwidth that spans over one octave around the resonances. It should be noted that the correlation between the measured and predicted absorption values of MCP2 degrade at and around the resonances, whereas an acceptable correlation is observed for the other materials. Indeed, Figure 2b shows that the channels of MCP2 are not exactly circular but rather hexagonal. Moreover, on a larger scale (not shown), it can be observed that the channels are grouped into hexagonal patches whose boundaries present some micro-porosity at the junctions, which is not accounted for in the model. These defects contribute to the discrepancies observed in Figure 8 for MCP2. The absorption performance of MCP2 is similar to that of the MPP resonant absorber, which produces a total absorption of 0.42 and a maximum absorption of 0.6 at the second resonance, but a narrower efficiency bandwidth (less than one octave).

Although the UMPP absorber achieves a mere total absorption of 0.51, it reaches 800 Hz, a near-unit absorption value with a resonance bandwidth of 1830 Hz, thereby achieving a so-called critically coupled condition [19] at a frequency for which all of the incident energy is dissipated by the UMPP. A second resonance can also be observed at 3650 Hz, reaching a high absorption value of 0.95 but with a narrow bandwidth of 450 Hz.

Regarding the unbacked case, Table 2 shows that MCP1 and UMPPs still appear to be the most performant materials when backed by a rigid cavity. The lowest frequency at which the MCP1 material achieves its highest absorption (0.92) is 942 Hz, making it a $\lambda/7$ sub-wavelength absorber at this frequency. At a frequency of 2400 Hz, at which the absorption of MCP1 stays above 0.8, it is still a $\lambda/2.5$ sub-wavelength absorber. By comparison, the UMPP is a $\lambda/8$ resonance absorber when it fully dissipates the incident energy at 800 Hz, albeit over a narrower bandwidth than MCP1.

Because the design of thin resonance absorbers that are efficient over a broad bandwidth is a current challenge, it is of interest to examine how the absorption properties of MCP1 are modified when decreasing the cavity depth from 55 mm down to 10 mm. The experimental and simulated absorption results are shown in Figure 9 with an acceptable correlation between the measured and modeled performance. The performance is summarized in Table 2.

As the cavity depth $D$ decreases, the equivalent stiffness of the air cavity increases so that the cut-off frequency, below which the absorption of MCP1 drops below 0.5, increases from 365 Hz to 910 Hz. This results in an upshift by 3500 Hz of the first resonance frequency, therefore turning a $\lambda/4$ resonance absorber at 1500 Hz ($\alpha_{max} = 0.92$, $D = 55$ mm) into a $\lambda/7$ sub-wavelength absorber at 5000 Hz ($\alpha_{max} = 0.82$, $D = 10$ mm). Moreover, the half-bandwidth of absorption is significantly increased from 2700 Hz for $D = 55$ mm up to 5800 Hz for $D = 10$ mm, with a steady value of the total absorption (0.69–0.76).

Figure 10 examines how the absorption properties of a UMPP are modified when decreasing the cavity depth from 55 mm to 20 mm, when compared to those of MCP1. This produces an upshift by 660 Hz of the first resonance frequency of the UMPP, therefore turning a $\lambda/8$ resonance absorber at 800 Hz into a $\lambda/11$ sub-wavelength absorber at 1500 Hz. When $D = 20$ mm, the first resonance peak still exhibits a near-unit absorption value, and its half-bandwidth increases from 1830 Hz to 2200 Hz. However, the full absorption property is lost if one further decreases the cavity depth beneath 20 mm. Like MCP1, the total absorption of the UMPP is almost unchanged (from 0.51 to 0.54) when decreasing the cavity depth. Interestingly, the efficiency ranges ($\alpha > 0.8$) of the UMPP and MCP1, backed by a 20 mm cavity, complete each other without overlapping (Figure 10, plain gray and

orange curves), covering ranges of 1–2 kHz for the UMPP and 2–6.5 kHz for MCP1. A parallel arrangement of these materials would lead to a compact ultra-broadband absorber.

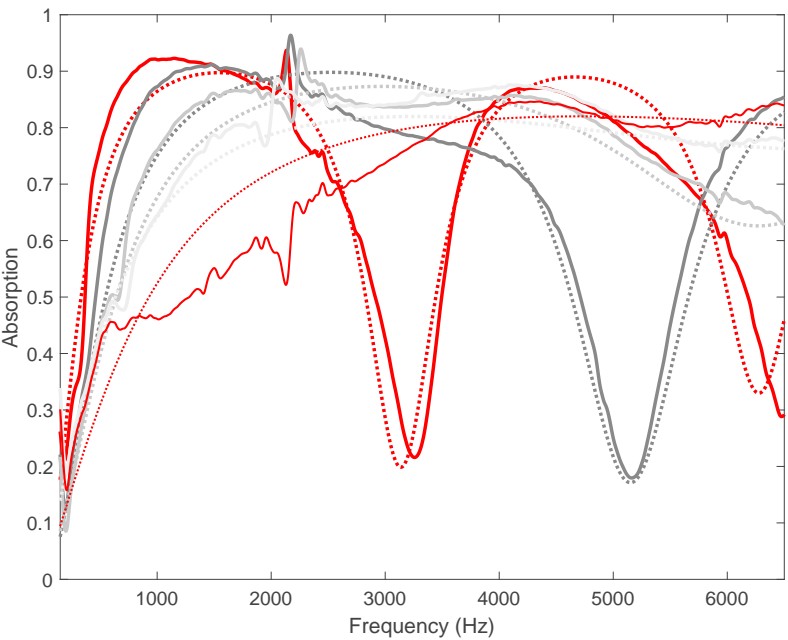

**Figure 9.** Simulated (dotted) and measured (plain) absorption properties of MCP1 when backed by a rigid cavity with depths of 55 mm (thick red), 35 mm (dark gray), 25 mm (gray), 20 mm (light gray) and 10 mm (thin red), assuming a normal incidence.

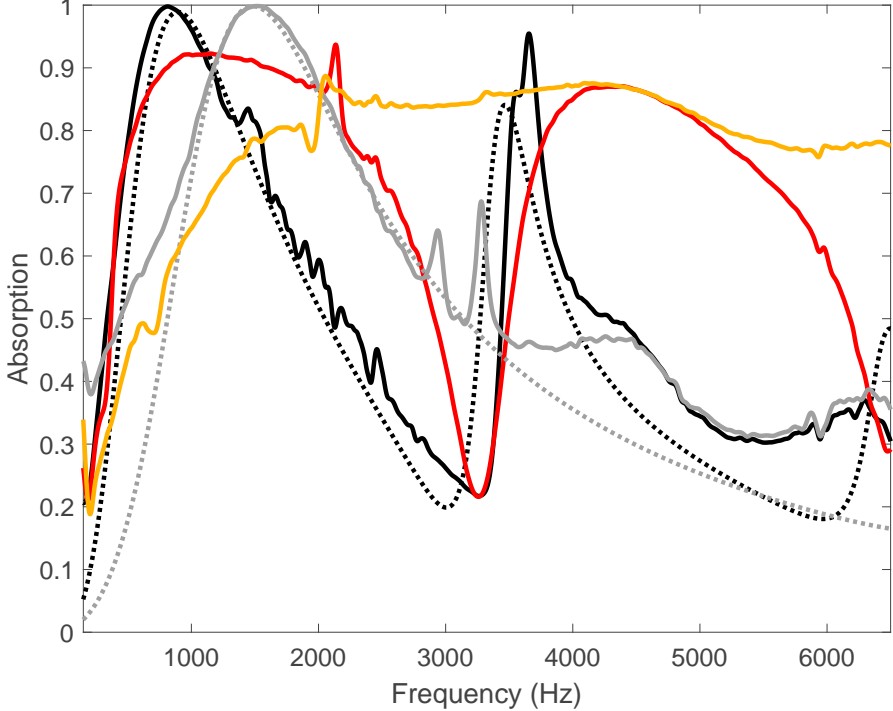

**Figure 10.** Simulated (dotted) and measured (plain) absorption properties of UMPP when backed by a rigid cavity with depths of 55 mm (thick black) and 20 mm (thick gray), under normal incidence; also shown are the measured absorption properties of MCP1 when backed by a rigid cavity with depths of 55 mm (thick red) and 20 mm (thick orange).

By removing the rigid bottom of the cavity, a stiffness-controlled closed cavity can be turned into a mass-controlled open cavity, which may potentially remove the absorption dip observed in Figures 8–10 at very low frequencies. This is verified in Figure 11 according to both simulations and experiments, with a good correlation between both types of results. The maxima of absorption are then caused by half-wavelength resonances with a maximum particle velocity over the materials' surface, whereas absorption dips are due to quarter-wavelength resonances with a low particle velocity over the materials.

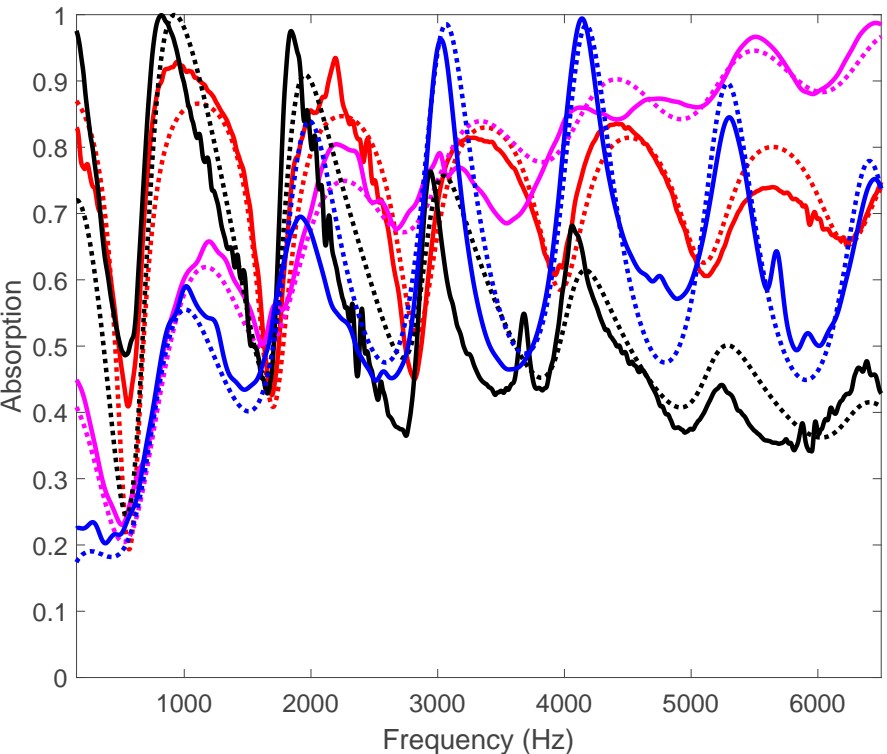

**Figure 11.** Simulated (dotted) and measured (plain) absorption properties of MCP1 (red), MCP2 (magenta), UMPP (black) and MPP (blue) backed by an open cavity with a depth of 152 mm, assuming normal incidence.

Below 1800 Hz, MCP1 and UMPP clearly outperform MCP2 and MPP, with high absorption values (greater than 0.8) observed at the lower frequency limit (80 Hz) for the Kundt tube measurements. This trend tends to reverse above 1800 Hz because the amplitudes of the resonances linked to MCP1 and the UMPP diminish as the frequency increases. This is due to an increasing leakage of the sound radiated by the open termination of the cavity with frequency, as shown in Figure 7. Meanwhile, the absorption linked to MCP2 steadily increases up to unity toward 6500Hz with wavy variations due to half-wavelength resonances. One of the absorption peaks associated with the MPP resonator even reaches near-unit values at 4100 Hz. The total absorption calculated over the whole frequency range is then dominated by MCP1 (0.73) and MCP2 (0.73) due to their high absorption performances at low and high frequencies, respectively. The UMPP and MPP still provide an acceptable overall performance with total absorptions of 0.55 and 0.58, respectively. Decreasing the depth of the open cavity (not shown) produces absorption results that progressively tend toward those shown in Figure 5 for the unbacked materials.

## 5. Discussion

The absorption performances of the different materials in backed or unbacked configurations are summarized in Table 3 with $\overline{\alpha}_{\text{unb.}}$, $\overline{\alpha}_{55}$ and $\overline{\alpha}_{20}$, the total absorptions calculated between 80 Hz and $f_{\text{cut-on}}$ for the unbacked case and for cases in which the materials are backed by a rigid cavity with depths of 55 mm and 20 mm, respectively. In each case, $\Delta\lambda/\overline{D}$

corresponds to the bandwidth-to-thickness ratio of the absorber, where $\overline{D}$ is the overall thickness of the absorber ($\overline{D} = t$ for the unbacked case and $\overline{D} = t + D \approx D$ for the backed case). $\Delta\lambda$ is defined as

$$\Delta\lambda = \lambda_{\max} - \lambda_{\min} = c_0\left(\frac{1}{f_{\min}} - \frac{1}{f_{\max}}\right) \tag{11}$$

**Table 3.** Overview of materials' absorption performance.

| Material | $\overline{\alpha}_{\text{unb.}}$ | $\Delta\lambda/\overline{D}\big|_{\text{unb.}}$ | $\overline{\alpha}_{55}$ | $\Delta\lambda/\overline{D}\big|_{55}$ | $\overline{\alpha}_{20}$ | $\Delta\lambda/\overline{D}\big|_{20}$ |
|---|---|---|---|---|---|---|
| MCP1 | **0.84** | **4150** | **0.71** | 6 | **0.75** | 8 |
| MCP2 | 0.58 | 26 | 0.45 | – | – | – |
| UMPP | **0.55** | **7890** | 0.51 | 5.5 | 0.54 | 7.6 |
| MPP | 0.47 | – | 0.42 | – | – | – |

The frequency bounds $f_{\min}$ and $f_{\max}$ exist such that the absorption exceeds 0.8 between these bounds. If the absorption stays greater than 0.8 above the first duct cut-on frequency $f_{\text{cut-on}}$, then $f_{\max}$ is equal to $f_{\text{cut-on}}$ because the total absorption is evaluated up to $f_{\text{cut-on}}$. If the absorption stays greater than 0.8 at very low frequencies, then $f_{\min}$ is equal to $f_{\text{low}}$ (80 Hz), the lowest frequency limit for the Kundt tube measurements. If the absorption stays below 0.8 up to $f_{\text{cut-on}}$, then $\Delta\lambda$ cannot be calculated. A possible indicator of the high broadband acoustic efficiency of the absorbers is given by large values of the index $\overline{\alpha}\,\Delta\lambda/\overline{D}$. This corresponds to the bold numbers in Table 3.

Regarding rigidly backed micro-perforated absorbers, the causality principle imposes an upper bound on their bandwidth-to-thickness ratio [20], given by

$$\frac{\Delta\lambda}{D} \leq \frac{4\pi^2}{|\log(1 - \alpha_0)|}, \tag{12}$$

stating that a large absorption $\alpha_0$ can only be achieved over a limited bandwidth $\Delta\lambda$ or that it requires a deep cavity. By choosing $\alpha_0 = 0.8$, Equation (12) leads to the ultimate bandwidth-to-thickness ratio, $\Delta\lambda/D \leq 24.5$. This is verified in Table 3 through the values of $\Delta\lambda/\overline{D}\big|_{55}$ or $\Delta\lambda/\overline{D}\big|_{20}$ not exceeding 24.5. Regarding unbacked absorbers, to the authors' knowledge, no similar criterion has yet been derived, and the ratio $\Delta\lambda/\overline{D}$ can reach quite high values, as seen by $\Delta\lambda/\overline{D}\big|_{\text{unb.}}$.

In Table 3, the highest acoustic efficiency for unbacked absorbers is reached by UMPP, being slightly above that of MCP1, due to the ability of the UMPP to reach similar absorption values compared to MCP1, albeit over a reduced bandwidth but with a thickness that is half less than that of MCP1. However, given the minute thicknesses involved, more weight should be given to $\overline{\alpha}$ and $\Delta\lambda$ in the unbacked case, with respect to $\overline{D}$. MCP1 then outperforms the UMPP with absorption values exceeding 0.8 over 5.5 octaves, leading to a total absorption of 0.84. This is due to a near-optimal channel radius, 14 μm, for MCP1. This leads to a specific transfer resistance, equal to 1.9 (Equation (5) and Figure 3a) over an ultra-broad frequency range. This value is nearly equal to the theoretical optimal resistance value for an unbacked MCP, equal to 2 [8]. Such resistance is achieved over a maximum bandwidth controlled by the Shear number, Sh, which should stay below 1 as a necessary condition for the VBL to occupy the full channel radius and efficiently dissipate the acoustic energy. This is the case for MCP1 because Sh < 0.73 up to $f_{\text{cut-on}}$, unlike the UMPP, for which Sh > 1 above 200 Hz. This explains the reduced efficiency bandwidth for the UMPP with respect to MCP1. Moreover, visco-thermal dissipation essentially occurs within the parallel micro-channels of MCP1 due to its high aspect ratio of 37:1. Conversely, significant contributions to the viscous dissipation arise from vortex shedding at the inlet/outlet of the apertures of the UMPP with high acoustic flow distortion induced by a large hole pitch. Regarding rigidly backed absorbers, the index $\overline{\alpha}\,\Delta\lambda/\overline{D}$ is well suited. Table 3 shows that, regardless of the cavity depth, the MCP1 resonant absorber has the highest acoustic

efficiency index amongst the materials, followed by the UMPP resonant absorber. As the cavity depth decreases, the bandwidth-to-thickness ratio, $\Delta\lambda/\overline{D}$, increases toward the ultimate one (24.5). Assessing how the MCP absorbers perform with respect to the existent solutions would be of interest.

A comparison is shown in Figure 12 between the performance of MCP1 absorbers as well as state-of-the-art broadband absorbers such as an anechoic termination composed of melamine foam [11] (Figure 12a), a metamaterial composed of a parallel array of coiled resonators, eventually covered by a sponge [5] (Figure 12b) and a rigidly backed MPP connected to hollow flexible tubes inserted into a rigid cavity [6] (Figure 12c).

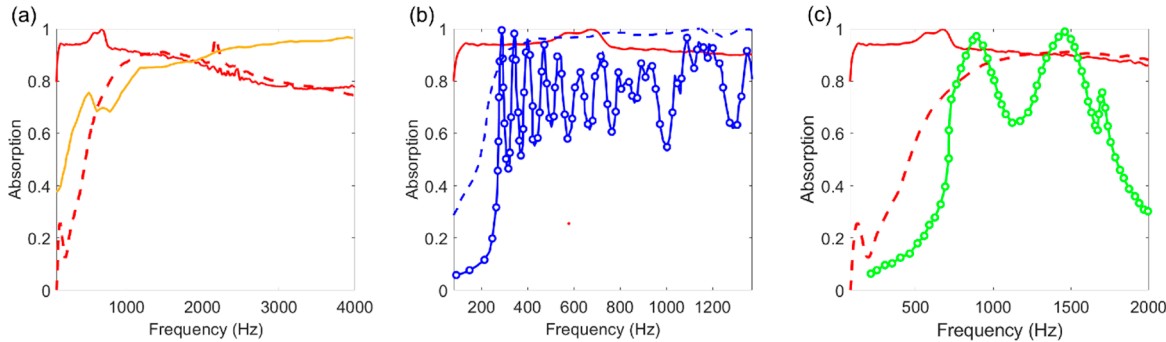

**Figure 12.** (**a**) Absorption properties of unbacked MCP1 (plain red), MCP1 backed by a 55 mm cavity depth (dashed red) and a rigidly backed 200 mm anechoic cone of melamine foam (orange, reproduced from Figure 2 in [11]); (**b**) Absorption properties of unbacked MCP1 (plain red), a 110.6 mm depth metamaterial composed of a parallel array of 4 × 4 quarter-wavelength coiled resonators (blue with circles, reproduced from Figure 2 in [5]) and the same metamaterial covered by a 10 mm sponge (dashed blue, reproduced from Figure 2 in [5]); (**c**) Absorption properties of unbacked MCP1 (plain red), MCP1 backed by a 35 mm cavity depth (dashed red) and an MPP connected to hollow tubes of variable length in a rigid backing cavity with a depth of 36 mm (green circles, reproduced from Figure 17 in [6]); all the curves are obtained from impedance tube measurements.

It can be seen from Figure 12a that unbacked MCP1 offers a solution for ultra-short low-frequency anechoic terminations for Kundt tube measurements, up to 2 kHz, complementary to current anechoic terminations composed of a 200 mm cone of melamine foam backed by a rigid end cap. Above 2 kHz, the melamine cone offers better absorption (0.96 at 4 kHz) than unbacked MCP1 (0.77 at 4 kHz) but has a lower bandwidth-to-thickness ratio $\Delta\lambda/\overline{D}\big|_{200} = 1.28$ and becomes reflective at low frequencies, with the absorption decaying from 0.7 toward zero below 400 Hz [11]. Instead, unbacked MCP1 produces a controllable amount of near-constant absorption above 0.87 over an ultra-broad bandwidth, typically 100 Hz–2 kHz. Figure 12a also shows that a rigidly backed MCP1 with $\Delta\lambda/\overline{D}\big|_{55} = 6$ exceeds the absorption of the anechoic termination by up to 15% between 600 Hz and 2 kHz but produces lower absorption below 600 Hz. Hence, another potential application of an MCP1 resonant absorber could be reducing the noise generated by fuel-efficient ultra-high bypass ratio turbofan engines in aeronautical transport with a targeted bandwidth of 600 Hz–2 kHz [21]. This would, however, require a specific study under grazing sound incidence in the presence of flow.

Figure 12b shows that unbacked MCP1 provides a better (resp. almost comparable) absorption performance below (resp. above) 250 Hz with respect to a metamaterial with a thickness of 110.6 mm composed of a parallel array of coiled quarter-wavelength cavity resonators [5] whose individual anti-resonances generate several absorption dips. Covering the metamaterial cavity mouths with a suitable amount of resistive porous material smooths out the absorption dips and provides a near-constant absorption that is greater than 0.9 over a broad bandwidth of 400–1370 Hz [5]. Unbacked MCP1 provides similar a performance between 400 Hz and 700 Hz with, however, lower constant absorption values (between 0.9 and 0.97) above 700 Hz, whereas they exceed 0.97 with the enhanced metamaterial with a

bandwidth-to-thickness ratio of $\Delta\lambda/\overline{D}\big|_{120.6} = 9.2$. Between 80 Hz and 250 Hz, unbacked MCP1 clearly overcomes the drop in absorption observed with the enhanced metamaterial.

Figure 12c compares the absorption performance of a 36 mm MCP1 resonator to that due to an MPP (1 mm thick, perforation ratio 3.9%) connected to flexible hollow tubes of two variable lengths (10 mm and 20 mm), located above a partitioned 35 mm rigid cavity [6], a so-called MPP-T. The MCP1 resonator is efficient over a much broader frequency range with a half-bandwidth of 1560 Hz, instead being 330 Hz for MPP-T, with almost a constant absorption value of 0.88 above 1 kHz, whereas the MPP-T reaches near-unit absorption at 870 Hz and 1460 Hz. At low frequencies below 760 Hz, the MCP1 resonator provides a greater amount of absorption compared to the MPP-T (by up to 40%) due to its broad bandwidth. If used as a liner in aeronautical applications, MCP1 would therefore be better suited to mitigate broadband noise components, whereas the MPP-T is designed to target specific tonal components.

The downside of the above-mentioned rigidly backed absorbers (such as the MCP1 absorber, the cone anechoic termination or the enhanced metamaterial) is their drop in performance as the frequency decreases due to the increased stiffness of the air cavity. A solution would be either to use an unbacked MCP1, which does not necessarily provide the best performance above 700 Hz, or a parallel arrangement between MCP1s, in which part of it would be backed by a shallow cavity (with a depth of 10 mm to bring a near-constant absorption of 0.85 over 2–6.5 kHz; see Figure 10), and part of it would be unbacked (to bring a similar amount of absorption over 100 Hz–2 kHz; see Figure 5). A more robust structural design would be to load the latter part by a shallow open cavity of the same depth as the rigid one. The ultra-broadband performance of this sub-wavelength parallel MCP absorber works according to two resonant mechanisms: the first half- (resp. quarter-) wavelength resonance for the open (resp. closed) backing cavity contributes to achieving high particle velocities through the micro-channels of MCP1. An optimization study would then be required to ensure perfect continuity and low overlap between the absorption performance of the individual resonators.

## 6. Conclusions

This theoretical and experimental study assesses the normal incidence absorption performance of highly porous micro-capillary plates with micron-sized channels against those of more standard micro-perforated panel absorbers with sub-millimetric-sized holes. MCPs are modeled with pure resistances calibrated according to their channel diameter, thickness and perforation ratio. This model is verified using impedance tube measurements. It is found that unbacked 1 mm-thick MCPs with both a high porosity (greater than 60%) and channels with a diameter of 28 μm are able to provide absorption values greater than 0.87 above a broad bandwidth, typically 100 Hz–2 kHz. Increasing the diameter while keeping high (resp. low) porosity drastically lowers the efficiency of the bandwidth (resp. the maximum absorption amplitude) that can be reached.

Given their very large bandwidth-to-thickness ratio, unbacked MCPs could be used in a quiet environment as thin low-frequency anechoic terminations, complementary to existing anechoic cone terminations, that are more efficient at higher frequencies. It would be of interest to determine how robust the unbacked MCP performance is for the reactivity index of the room into which it radiates.

This study also shows that MCP resonators backed by rigid shallow cavities have ultra-broadband absorption properties. They are found to have the highest acoustic efficiency index with respect to other resonating absorbers with either a larger aperture diameter or a lower perforation ratio (such as standard MPPs), but also with respect to engineered metamaterial absorbers. MCP resonators could potentially be used as compact lining solutions to tackle the low-mid broadband frequency content of turbofan noise in aeronautical applications, although this warrants dedicated aero-acoustic studies. A parallel MCP resonant absorber partly backed by closed and open shallow cavities is proposed to overcome the limited performance at very low frequencies of resonating absorbers while

maintaining high broadband performance at mid/high frequencies. This hybrid-type MCP absorber would require further studies.

**Author Contributions:** Conceptualization, T.B. and C.M.; methodology, C.M. and T.B.; software, C.M.; validation, T.B.; formal analysis, C.M. and T.B.; investigation, T.B. and C.M.; resources, T.B. and C.M.; data curation, T.B.; writing—original draft preparation, T.B. and C.M.; writing—review and editing, C.M.; visualization, C.M.; supervision, T.B.; project administration, T.B. and C.M.; funding acquisition, T.B. and C.M. All authors have read and agreed to the published version of the manuscript.

**Funding:** This work is part of the project TED2021-130103B-I00, funded by MCIN/AEI/10.13039/ 501100011033 and the European Union "NextGenerationEU"/PRTR. It also received support from the French government under the France 2030 investment plan, as part of the Initiative d'Excellence d'Aix-Marseille Université-A*MIDEX (AMX-19-IET-010).

**Institutional Review Board Statement:** Not applicable.

**Informed Consent Statement:** Not applicable.

**Data Availability Statement:** Data supporting the reported results are available on request.

**Conflicts of Interest:** The authors declare no conflict of interest.

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
