# Peer review of "From Micro-Perforates to Micro-Capillary Absorbers: Analysis of Their Broadband Absorption Performance through Modeling and Experiments"

_applsci, doi:10.3390/app131910844_

Round 1

Reviewer 1 Report

The article "From micro-perforates to micro-capillary absorbers: analysis of their broadband absorption performance through modelings and experiments" is well organized and shows merit in the field of research. Need minor revision to process.

1)   MCP resonant absorber range need to be mentioned and provide the potential applications at low frequency band. 

2) Absorption performance at the individual resonators need to eloborate

3) Comparative analysis before the conclusion with past literature will project the novelty of the work. 

NA

Author Response

Please, see document attached

Reviewer 2 Report

Concerning the manuscript titled “From micro-perforates to micro-capillary absorbers analysis of their broadband absorption performance through modelling and experiments,” reviewer would like the authors to clear the following items:

(i) The use of both simulations and experiments for what purpose? To verify the simulation results or to investigate the proposed structure?

(ii) Explanation for the lack of Conclusion section.

(iii) Reasons for choosing the parameters of micro-porous panels as shown in Table 1.

(iv) Add scale bar for Figures 1(a), 1(b), 1(c), 1(d) and Figures 2(b), 2(c).

(v) What are the differences between Figures 3(a) and 3(b)?

(vi) In line 217, what does Z1 stand for?

(vii) Comparison on performance between the author’s work and the existent publications?

Decisions: Depending on the response from the authors, the manuscript will be accepted to publish or not.

The English of the manuscript is acceptance.

Author Response

Please, see document attached

Round 2

Reviewer 2 Report

The revised manuscript has been improved and meets the required quality and impact of the Journal.

It is suggested to accept the manuscript in this form for publication in Applied Sciences.

The authors should check the manuscript again to avoid the typing errors and improve the English.